# Child public health indicators for fragile, conflict-affected, and vulnerable settings: A scoping review

**Ayesha Kadir** [1], **Amy J. Stevens**[2], **Emi A. Takahashi**[3], **Sham Lal**[4]*

**1** Save the Children International, London, United Kingdom, **2** Yorkshire and Humber School of Public Health, Leeds, United Kingdom, **3** Independent Consultant in Epidemiology, Santa Barbara, California, USA, **4** Faculty of Infectious Tropical Diseases, London School of Hygiene and Tropical Medicine, London, United Kingdom

* sham.lal@lshtm.ac.uk

## Abstract

Children and young people are disproportionately vulnerable to harm during crises, yet child public health expertise is limited in humanitarian settings and outcomes and impact data are lacking. This review characterises child public health indicators that are routinely collected, required by donors, and recommended for use in fragile, conflict-affected, and vulnerable (FCV) settings. We conducted database and grey literature searches and collected indicators from technical agencies, partnerships, donors, and nongovernmental organisations providing child public health services in FCV settings. Indicators were included if they were child-specific or disaggregated for ≤18 years. Indicators were coded into domains of health status, health service, social determinants, and health behaviours and analysed for trends in thematic focus and clarity. A total of 668 indicators were included. Routinely collected indicators (N = 152) focused on health status and health services. Donors required only 14 indicators. Technical bodies and academics recommended 502 indicators for routine measurement. Prioritised topics included nutrition, paediatrics, infectious diseases, mortality, and maternal-newborn care. There were notable gaps in indicators for child development and disability. Child protection indicators were not routinely collected, despite being the focus of 39% of recommended indicators. There were overlaps and duplications, varied age disaggregations, and 49% of indicators required interpretation to measure. The review demonstrates that it is feasible to routinely measure child public health outcomes in FCV settings. Recommendations from technical agencies and partnerships are characterised by numerous indicators with duplication, poor definitions, and siloed sector-specific focus. There are gaps in measurement of critical child public health topics. To improve safety and effectiveness of interventions for child public health, consensus is needed on priority topics and a shortlist of quality, standardised indicators that governmental and nongovernmental actors can be reasonably expected to measure. Indicators should be prioritised to support decision-making and include proxy indicators for periods when routine measurement is hampered.

**Data availability statement:** The full protocol and anonymized full dataset are available in the supporting information.

**Funding:** The authors received no specific funding for this work.

**Competing interests:** The authors have declared that no competing interests exist.

## Introduction

Humanitarian crises present critical threats to human health and security. Armed conflict, climate-related disasters, hunger, forced displacement, and communicable disease outbreaks are associated with the disruption and collapse of essential health services, food supply chains, water and sanitation facilities, education, and social protection programmes. Children and young people are disproportionately vulnerable to direct and indirect harms associated with crises, with nearly 1 in 5 living in or fleeing from armed conflict [1]. Child mortality data in crisis contexts are lacking [2], however the available data suggest that under 5 mortality rates in fragile and conflict-affected countries is almost three times higher than in countries not classified as fragile or conflict-affected, and nearly twice that of all countries classified as low- or middle income [3].

Few crises happen in a vacuum, however, and increasingly, crises are protracted. The result is that children face multiple forms of adversity before the crisis occurs, throughout the crisis, and often long after the acute phase has subsided. The harmful effects of early childhood adversity have been well documented in high income settings, with immediate and long-term impacts on physical health; mental health; growth; cognitive, social and emotional development; and life trajectories [4]. Interventions to protect and promote child health in FCV settings should be grounded in an understanding of the pervasive impact of adversity on children; this requires collaboration with sectors not traditionally considered to fall within the domain of public health, such as child protection and education. Additionally, the measurement of outcomes and impacts relating to severe adversity on child health and development is needed to adequately inform and monitor the safety and effectiveness of child public health interventions.

Armed conflict and humanitarian disasters often occur in settings where weak public health information systems become further strained by crisis. In most fragile, conflict-affected, and vulnerable (FCV) settings, there is limited data on the health needs of children. In the absence of routine child public health data, humanitarian actors delivering public health services in these settings often turn to survey data. The Demographic and Health Survey (DHS) and/or the Multiple Index Cluster Survey (MICS) may be used to support decision-making and prioritisation of interventions. These surveys capture public health data from only a single point in time and may be out-dated by the time they are used to inform an intervention. During humanitarian responses, the existing survey information is usually triangulated with a rapid needs assessment and the available governmental data [5, 6]. In the end, however, many humanitarian interventions are developed and delivered without having up-to-date child public health data to inform them. At the same time, nongovernmental actors working in FCV settings are actively collecting a large amount of data for internal reporting, donor reporting, and reporting to governments; so much data are collected that the United Nations Office for Coordination of Humanitarian Affairs (OCHA) established a Centre for Humanitarian Data to improve data use and data practices [7]. There is a huge opportunity for the improved measurement and use of child public health data to inform interventions and to monitor and evaluate outcomes.

The need for standardised, high-quality child public health data in FCV settings has never been greater. Recent years have seen a growth in the need for assistance to crisis-affected populations and simultaneous reduction in funding to support humanitarian response. As of 30 April 2024, the Inter-Agency coordinated funding appeals totalled US$48.2 billion, however the reported received funding is only 15% of this - the lowest level of funding in the same period since 2019, at $1.39 billion [8]. Simultaneously, monitoring and understanding the effectiveness of humanitarian aid is difficult due to a lack of standardised performance indicators disaggregated by age and gender, and in particular, lack of measurement of outcomes

[9]. Increased need with simultaneously reduced funding makes it even more important to base operational decisions on sound, up-to-date, and relevant information. The effectiveness of interventions to support children in humanitarian contexts depends on trustworthy and appropriately detailed data on child health status, health risks, and services available before the crisis hit as well as during the crisis and after the acute phase has passed. These data are essential to allocate scare funds effectively, establish priorities, evaluate the safety and performance of interventions, and maintain accountability to affected populations.

Current challenges to understanding the public health needs of children in FCV settings include a lack of consensus on the definition of a child, intermittent data collection, wide variations in disaggregation of indicators by age, limitations in public health infrastructure in humanitarian settings, siloed interventions by multiple sectors engaging in humanitarian response, and limited understanding of child public health by generalist public health actors [10]. These challenges to the collection of child public health data mirror challenges in delivering appropriate and effective child public health interventions in FCV settings.

A standard set of relevant, realistically obtainable core child public health indicators in fragile, conflict-affected, and vulnerable (FCV) settings is needed to improve our understanding of children's needs, support prioritisation of activities, inform interventions, support coordination of interventions and accountability, and track progress. This scoping review aims to make a step towards the development of a core set of child public indicators by providing an overview of the current indicator landscape and identifying overlaps, divergences, gaps, and potential for harmonisation and pooling of data.

## Methods

### Search strategy

This review characterises child public health indicators that are routinely collected, required, or recommended for use in FCV settings (Table 1). We use the term "humanitarian" to refer to crisis settings where external support is required to meet the needs of the affected population. The terms "crisis" and "humanitarian" are used interchangeably. The term "FCV" includes a broader range of settings, including pre- and post-crisis as well as protracted crises, for which external support may or may not be required to meet the population's needs. For the purpose of this review, child public health indicators were defined as summary measures that captured information on different attributes of children's health. We defined four child public health indicator domains for this review: health status, healthcare, social determinants of health, and health behaviours (See Table 1 for definitions and Table 2 for domains). These domains are based on the public health practice areas of health protection, health improvement, and health service delivery, considering the types of interventions conducted in FCV settings and the data required for their planning, monitoring, and evaluation. In order to identify indicators that are feasible to collect, required, and recommended for use in FCV settings, we collected lists of (i) indicators that are routinely collected by nongovernmental agencies providing public health services in FCV settings; (ii) required indicators from a sample of donors; and (iii) indicators recommended for routine use by technical agencies, technical advisory groups, partnerships, and academics working in public health in FCV settings.

There were two rounds of data collection in this review: 1) a search of the peer-reviewed and grey literature, and 2) collection of routinely collected, required, and recommended child public health indicators from operational agencies delivering child public health services in FCV settings, donors, and groups or organisations providing technical guidance for child public health programming, respectively.

**Table 1. List of definitions.**

| | |
|---|---|
| *Child* | **Any human being under the age of 18 years. In this review, younger children refers to children up to 10 years of age and adolescents refers to children aged 10-18 – the age 10 cut-off being consistent with WHO definition of an adolescent which is 10-19 years [11].** |
| *Child public health indicators* | Summary measures that capture information on different attributes of children's health status, the social determinants of child health, healthcare in relation to the child population and behaviours that influence child health. |
| *Fragile, conflict-affected and vulnerable (FCV) settings* | As per WHO usage, this term subjectively describes a range of situations including humanitarian crises, protracted emergencies, and armed conflicts. In general, FCV settings experience disruption of routine health service organization and delivery systems, increased health needs, complex resourcing landscapes, and vulnerability to further public health crises [12]. External assistance may or may not be required to meet the needs of the affected population. In protracted crises and in the period before and after acute crisis, access to the affected population may be more reliable, and infrastructure may be more functional than in humanitarian settings. |
| *Humanitarian settings* | Settings in which an event or a series of events has resulted in a critical threat to the health, safety, security, and well-being of the affected population. Examples include political unrest, armed conflict, natural disaster, extreme weather event, epidemic, or hunger crisis. The coping capacity of the affected community is overwhelmed, and external assistance is required [13]. |
| *Impact indicator* | A measure of the short and medium to long-term results of an intervention. This scoping review curated health outcome and response outcome indicators. |
| *Outcome indicator* | A measure of the change that has occurred as a result of an intervention. This can be at the individual level or in relation to a population-level intervention. |
| *Recommended indicators* | Indicators recommended for routine collection in FCV settings by agencies such as the World Health Organization or the Office for Coordination of Humanitarian Affairs; technical groups and partnerships, such as the Alliance for Child Protection in Humanitarian Action or the Healthy Newborn Network; and academics. |
| *Required indicators* | Indicators required by donors for reporting purposes. |
| *Routine Indicators* | Indicators that are collected and reported on a regular basis by actors delivering public health interventions in FCV settings. |
| *Situation indicators* | A measure of the current situation (e.g., baseline indicators) or describe what is required in crisis-affected areas (e.g., needs indicators). |

**Table 2. Child public health indicator domains.**

| | |
|---|---|
| *Health status* | Health status refers to states of good health and normal development, as well as morbidity and mortality related to physical and mental health; disease outbreaks; immunisation status; nutrition; disability; cognitive, psychomotor, and social development; and psychosocial wellbeing in the child population. |
| *Healthcare* | Healthcare encompasses both access to and quality of health service provision to children. Access barriers may be financial (direct or indirect), geographical, cultural, related to language or literacy, logistical, security-related, or associated with individual or institutional discrimination. |
| *Social determinants of health* | "The conditions in which people are born, grow, work, live, and age, and the wider set of forces and systems shaping the conditions of daily life"[14]. The social determinants of health in fragile, conflict-affected, and vulnerable settings are broad but key areas relevant to child health have been identified based on humanitarian response clusters. These include: child safety and protection; education and school attendance; food security; water, sanitation and hygiene (WASH); and shelter. Weakening of familial or communal protective mechanisms is a recognised risk factor for increased vulnerability to adverse health outcomes in children in FCV settings [15]. Children in FCV settings are at increased risk of experiencing death of a parent, separation from their caregivers, family and/or communities, and having a caregiver who is physically or mentally ill or disabled; therefore any indicators covering this area were also included. |
| *Health behaviours* | Intentional or unintentional actions taken by individuals that affect health or mortality either negatively or positively [16]. Examples include smoking, substance use, diet, physical activity, sleep, risky sexual activities, healthcare seeking behaviours, and compliance with medical advice/treatments. Health behaviours may be influenced by the social determinants of health and by the social norms people are exposed to, both of which may be adversely impacted in times of humanitarian crises. Health behaviours of children themselves, their caregivers and community members can all impact child population health. |

## Peer-reviewed and grey literature database searches

Database searches were developed by AS with support from AK and an experienced librarian at the Leeds Teaching Hospitals NHS Trust and undertaken 14-24 March 2024. AS conducted searches in Embase and MEDLINE from 1st January 2013 to 24th March 2023, with English language restrictions. The searches included variations of terms for: (1) fragile, conflict-affected, and vulnerable settings; (2) child; (3) public health; (4) indicators. The final search strategy for EMBASE is available in S1 Text. Identified records were exported into the online reference manager, Rayyan [17]. Duplicates were removed and the records were screened by AS and AK.

A grey literature search was run by AS in DuckDuckGo [18] and through screening the websites of major agencies supporting child public health activities in FCV settings. The DuckDuckGo search included the terms 'Child public health indicators humanitarian' and expanded to include 'all regions', 'anytime', in a 'moderate safe search'. Organisational websites searched included the World Health Organisation (WHO), WHO Global Health Cluster (GHC), United Nations Office for the Coordination of Humanitarian Affairs (OCHA), United Nations Children's Fund (UNICEF), United Nations High Commission for Refugees (UNHCR), Sphere Handbook, United States Agency for International Development (USAID), European Civil Protection and Humanitarian Aid Operations (ECHO), the United Kingdom Foreign, Commonwealth and Development Office (FCDO), Alliance for Child Protection in Humanitarian Action, Inter-agency Network for Education in Emergencies, Healthy Newborn Network, Inter-Agency Working Group on Reproductive Health in Crises, Interagency Gender Working Group, and ReliefWeb.

Reference lists of identified records from the database and grey literature searches were scanned and relevant references were reviewed to identify additional documents containing routinely collected, required, or recommended child public health indicators for FCV settings. AS and AK independently screened all records obtained by the database and grey literature searches. Disagreements were resolved by discussion. Records from the peer-reviewed literature and grey literature searches were included for extraction of indicators if they contained at least one indicator that met the inclusion criteria (Table 3).

## Searches for indicators from operational agencies, donors and technical advisory bodies

AK and AS contacted representatives of donors and operational agencies working with children in FCV settings to identify documents detailing the child public health indicators that are either actively used, required, or recommended for routine collection. The organisations contacted included WHO, UNICEF, UNHCR, International Organisation for Migration (IOM), OCHA, Save the Children, Médecins Sans Frontières (MSF), International Committee of the Red Cross (ICRC), International Rescue Committee (IRC), Médecins du Monde, USAID, FCDO, ECHO, the Bill & Melinda Gates Foundation, and the focal points at one operational agency responsible for reporting to the Danish International Development Agency (DANIDA) and the Lego Foundation, respectively.

## Screening of indicators

Indicator inclusion criteria were pilot tested by AK and AS and refined by the entire author group. The indicators contained in the documents from the database and grey literature searches as well as all indicators from operational agencies, donors and technical advisory bodies were extracted and compiled in Microsoft Excel [19]. All indicators were independently screened by AS and AK for inclusion in the review. Disagreements were resolved by discussion. When consensus was not reached or uncertainty persisted after discussion, disagreements were resolved by ET and SL in purpose-specific meetings attended by all authors, where the indicator(s) were presented and discussed, and consensus was reached.

Indicators were included if they were: quantitative; situation, outcome, or impact indicators (See Table 1. Definitions); child-focused; used or recommended for use in FCV settings, relevant to at least one of the defined domains of health status, health service, social determinants of health, and health behaviours; and contained a title and either a definition or description of calculation. Indicators included in the review were restricted to the born child. It is important to note that both maternal and other caregiver health and wellbeing have profound

**Table 3. Inclusion and exclusion eligibility criteria.**

Inclusion criteria
- Child-focused: pertaining to children 0-18 years or a disaggregated child age group in this bracket.
- Used, required, or recommended for routine use in FCV settings as defined in Table 1.
- Relevant to at least one of the defined child public health domains – health status, health service, social determinants of health, and health behaviours.
- Contain data and metadata (title, definition and/or description of how the indicator is calculated).
- Situation, outcome, or impact indicators.

Exclusion criteria
- Does not specifically mention newborns, infants, children, or adolescents AND does not disaggregate for at least one group <19 years old.
- Not specific to FCV settings or a subgroup of these settings (e.g., humanitarian settings)
- No definition or description provided
- Indicator not routinely collected or recommended/required for routine collection
- Input, process or output indicators
- Counts (e.g., "number of consultations")

impacts on child public health outcomes. However, in order to limit the size and scope of this review, maternal and caregiver health indicators were excluded. Indicators associated with maternal or caregiver health were only included if they specified age disaggregation of the caregiver such that caregivers and/or mothers ≤18 years old were visible in the disaggregation. (See Table 3 for inclusion and exclusion criteria, and S1 Text for the full protocol).

## Data analysis

A codebook was developed, piloted, and refined by AS, AK and ET. Coding and analysis was performed by AK. Included indicators were compiled in a Microsoft Excel Spreadsheet and coded thematically for relevance to the four domains and for clarity of the indicator definition (S2 Text). The final coded dataset was grouped into routinely measured indicators, required indicators for donors, and recommended indicators. The indicators in each of these groups were analysed respectively for trends related to measurement, theme, service type, service quality and/or accessibility, age disaggregation, and clarity. Clarity was determined by the degree of interpretation required to measure the indicator (none, some, or significant interpretation required). See (S3 Text) for the PRISMA Scoping Review checklist.

## Results

The database searches identified 662 records. After removing duplicates, the titles and abstracts of 504 records were screened for eligibility. Seven records were screened in full text, of which one report contained indicators that were relevant for inclusion in the review. A further 135 further records were identified from grey literature searches, snowball searches, and indicator lists obtained from a sample of donors, operational agencies, and technical experts, of which 23 records contained indicators relevant for the review. The final dataset contained indicators from 24 records: three peer-reviewed publications, nine indicator lists from 8 global technical organisations or partnerships (OCHA, UNHCR, Sphere, Alliance for Child Protection in Humanitarian Action, Inter-Agency Working Group on Reproductive Health in Crises, Healthy Newborn Network, Inter-agency Network for Education in Emergencies and the Interagency Gender Working Group), three indicator lists from two donor agencies (Bureau for Humanitarian Assistance (USAID BHA) and ECHO), and nine indicator lists from five operational agencies (Save the Children, MSF, ICRC, IRC, and UNICEF). The indicators from the included records were extracted and screened to identify recommended, required, and routinely collected child public health indicators in FCV settings, respectively (Fig 1 and S4 Text).

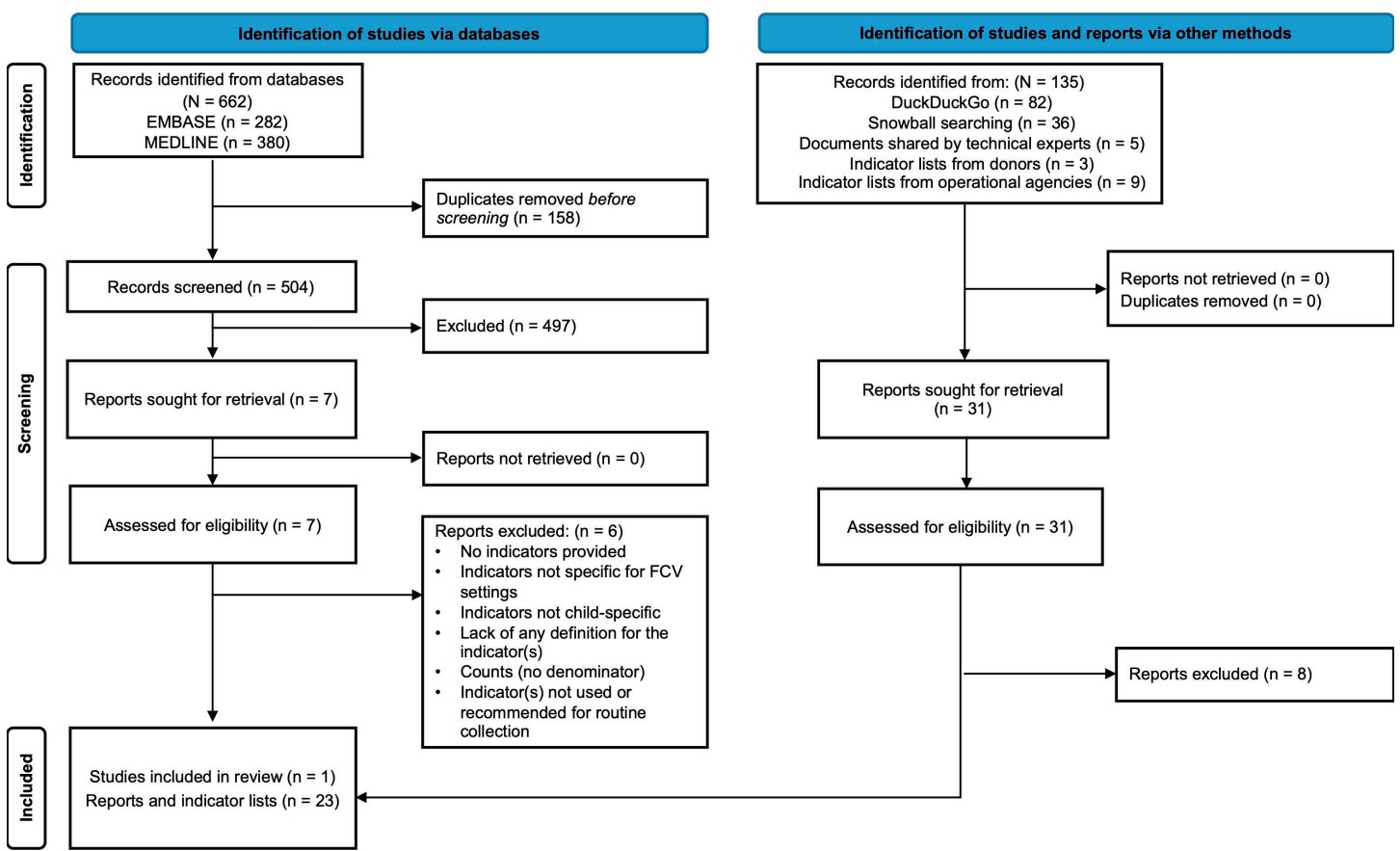

**Fig 1. Flow diagram.**

A total of 2784 indicators were extracted and screened, of which 668 met inclusion criteria (Table 4). The included indicators were categorised into (i) routinely collected; (ii) required by donors; and (iii) recommended for routine collection (see Table 1. Definitions, and Table 5. Included indicators by category) Four of the six donor agencies that were approached did not require specific indicators for reporting. The complete list of included indicators is available in supplementary materials (S1 Data).

Table 6 shows the number of indicators for each domain and Table 7 provides an overview of the thematic focus of the included indicators. As the tables show, the review identified a large number of indicators measuring a broad range of aspects of child public health, with overlaps across domains and thematic focus. Operational agencies mainly measured child health status and health services for children. Nearly two-thirds of routinely collected indicators measure quality or access to care. The two donors who required indicators asked for measures that were spread more evenly across the four domains. Recommending technical agencies, partnerships, and academics placed comparable focus on health status, health services, and social determinants, and less focus on health behaviours. Nutrition indicators were particularly numerous among all categories, comprising 43% of required, 22% of routinely measured, and 23% of all recommended indicators, respectively. Child protection indicators were numerous among recommended indicators (N= 197; 39%) but notably absent among routinely collected or required child public health indicators. One of the required indicators measured the general safety of service delivery but was not specific to child protection.

**Table 4. Indicators screened and included, by category.**

| Indicator category | Screened | Included |
|---|---|---|
| Routinely collected by operational agencies | 1088 | 152 (14%) |
| Required by donors | 276 | 14 (5.1%) |
| Recommended for routine collection | 1420 | 502 (35.4%) |
| Total | **2784** | **668 (24%)** |

**Table 5. Included indicators by category.**

| Indicator category | Total number of indicators | Number of indicators that did not require any interpretation* |
|---|---|---|
| Routinely collected by operational agencies | 152 | 85 (55.9%) |
| Required by donors | 14 | 8 (57.1%) |
| Recommended for routine collection | 502 | 231 (46.0%) |
| Total included indicators | **668** | **324 (48.5%)** |

*S2 Text.

**Table 6. Overview of included indicators by domain.**

| Domain | Number of indicators routinely collected by operational agencies (N=152) | Number of indicators required by donors (N=14) | Number of indicators recommended for routine collection (N=502) |
|---|---|---|---|
| Health Service | 136 (89.5%)) | 8 (57.1%) | 261 (52.0%) |
| Health Status | 124 (81.6%)) | 6 (42.9%) | 237 (47.2%) |
| Health Behaviours | 18 (11.8%) | 4 (28.6%) | 90 (17.9%) |
| Social Determinants | 21 (13.8%) | 5 (35.7%) | 240 (47.8%) |
| Total | **299** | **23** | **828** |

An individual indicator can measure more than one domain.

See Table 1 for definitions of routinely collected, required, and recommended and Table 2 for domain descriptions.

Similarly, 11% of recommended indicators measured an aspect of education, and 2 (14%) were required by donors, however education accounted for only 2 (1%) of routinely measured child public health indicators. Overall, 21 (14%) routinely collected indicators measured social determinants of child health. Agencies with an overt child rights mandate were more likely to measure aspects of child protection as well as social determinants of child health compared with other agencies.

There were overlaps and duplications among the indicators, as well as similar indicators with slight differences in definition that create challenges for pooling data. For example, there were a total of 61 child mortality indicators, which can be broadly grouped into measures of stillbirth (N = 4), newborn mortality (N = 10), infant mortality (N = 1), under-5 mortality (N = 6), adolescent mortality (N = 1), institutional mortality (N = 3), nutrition-related mortality (N = 12), other cause-specific mortality (N = 19), gendered mortality (N = 4), and a measure of the mortality review process (N = 1). One of the 61 mortality indicators was required by a donor, 31 were recommended for routine collection, and 29 were routinely collected by operational agencies – nearly one fifth of all routinely collected indicators measured an aspect

**Table 7. Overview of included child public health indicators by thematic focus[*].**

| Thematic focus | Routinely collected by operational agencies (N=152) | Required by donors (N=14) | Recommended by technical groups (N=502) |
|---|---|---|---|
| Nutrition | 34 (22.4%) | 6 (42.9%)[**] | 113 (22.5%) |
| Paediatrics | 32 (21.1%) | 0 (0%) | 72 (14.3%) |
| Infectious diseases | 31 (20.4%) | 0 (0%) | 27 (5.4%)[***] |
| Mortality | 29 (19.1%) | 1 (7.1%) | 31 (6.2%) |
| Continuum of maternal and newborn care | 28 (18.4%) | 3 (21.4%) | 93 (18.3%) |
| Social Determinants of Child Health | 21 (13.8%) | 0 (0%) | 240 (47.8%) |
| Adolescent Sexual and Reproductive Health | 15 (9.9%) | 0 (0%) | 24 (4.8%) |
| Immunisation and vaccination status | 9 (5.9%) | 0 (0%) | 13 (2.6%) |
| MHPSS | 7 (4.6%) | 0 (0%) | 20 (4.0%) |
| SGBV | 5 (3.3%) | 0 (0%) | 34 (6.8%) |
| Care-seeking behaviour | 4 (2.6%) | 0 (0%) | 7 (1.4%) |
| Water, Sanitation and Hygiene | 3 (2%) | 1 (7.1%) | 5 (1%) |
| Education | 2 (1.3%) | 2 (14.3%) | 55 (11%) |
| Noncommunicable disease | 2 (1.3%) | 0 (0%) | 3 (0.6%)[***] |
| Safety | 1 (0.7%) | 1 (7.1%) | 54 (10.8%) |
| Injury | 1 (0.7%) | 0 (0%) | 5 (1%) |
| Child protection | 0 (0%) | 0 (0%) | 197 (39.2%) |
| Integrated intersectoral indicators | 0 (0%) | 0 (0%) | 45 (9%) |
| Disability | 0 (0%) | 0 (0%) | 9 (1.8%) |
| Early child development | 0 (0%) | 0 (0%) | 0 (0%) |
| Quality | 69 (45.4%) | 4 (28.6%) | 129 (25.7%) |
| Access | 27 (17.8%) | 3 (21.4%) | 83 (16.5%) |
| Both quality and access | 0 (0%) | 1 (7.1%) | 19 (3.8%) |
| No interpretation required | 85 (55.9%) | 8 (57.1%) | 231 (46%) |
| Some interpretation required | 56 (36.8%) | 4 (28.6%) | 227 (45.2%) |
| Significant interpretation required | 11 (7.2%) | 2 (14.3%) | 44 (8.8%) |
| Facility-based health services | 44 (28.9%) | 4 (28.6%) | 97 (19.3%) |
| Community-based health services | 3 (2%) | 0 (0%) | 6 (1.2%) |

[*]An individual indicator can measure more than one of the listed thematic areas of focus.

[**]Includes Food Security and Livelihoods

[***]Does not include mortality indicators

of child mortality. Stillbirths were variously defined as "including stillbirths with no foetal heartbeat at admission as well as other stillbirths", "number of foetuses and infants born with no sign of life and born with birthweight of 1000g or more, or after 28 weeks gestation, or 35cm or more body length in a specified time period", "number of foetuses and newborns born after 28 weeks gestation or ≥ 1000g with no sign of life", and "foetal death after 22 weeks of gestation and prior to delivery". Newborns were again defined variably, some indicators without a definition of the newborn and others providing definitions of <28 days, or ≤28 days, or "the first 28 days of life". Some of the indicators measured institutional mortality while others measured population-level mortality. Some indicators were specific to a period of time while others were not. Nutrition mortality indicators similarly had overlaps and variations in

measures including inpatient and outpatient, severe acute malnutrition and moderate acute malnutrition, and supplementary feeding programme deaths. The definitions were again varied and some indicators were not defined. Age disaggregations for the nutrition mortality indicators appeared to include a typo as well as variations, including no disaggregation, <5, <5 years and ≥5 years, "6-59 months, 6-23 months, 24-59 months", and "0-5 months, 6-23 months, 24-59 months". The other mortality indicators had similar variations in definition as well as overlaps. Table 8 provides an overview of mortality indicators.

The pattern of similar indicators with slightly different definitions was also seen with other measures, most notably the continuum of maternal and newborn care, nutrition, and infectious diseases (S1 Data). These were also more commonly recommended and measured thematic areas. Two of the five operational agencies routinely measured under-five mortality, neonatal mortality, and stillbirth rates. As with the mortality indicators, there was significant heterogeneity in age disaggregation for all other indicators included in the review and for the definitions of children at varying ages and stages of life.

None of the operational agencies who shared indicators included measures of early child development (ECD) or disability outcomes in children, nor did the review identify recommended or required indicators for these topics. Among the recommended indicators, only agencies focusing on child protection or education recommended measurements of education (N = 55) or access to services for children with physical impairments or disabilities (N = 12). There were no recommended measures of ECD by any international organisation or partnership, regardless of the organisation/partnership thematic focus (i.e., health, public health, protection, education, or child rights). Health and public health agencies and partnerships did not recommend measurement of disability outcomes in children. There were, however, 14 indicators for which a disaggregation by disability status was recommended (S1 Data).

Agencies who provided routinely measured indicators in FCV settings placed a heavy focus on outcomes for mortality, nutrition, the continuum of maternal and newborn care, and infectious diseases (Table 7). A broad range of indicators for clinical child health services that did not fall into one of the other categories was coded as "paediatrics"; these paediatrics indicators account for one-fifth of the routinely measured child public health outcome indicators. There was also a heavy focus on adolescent sexual and reproductive health (n = 15, 9.9%), and broadly in the social determinants of child health (n = 21, 13.8%), again with a notable lack of measurement of ECD or disability outcomes. The two non-communicable disease (NCD) indicators that were routinely measured are most relevant for adults (unspecified type of diabetes, and hypertension), however, because they included age disaggregation for children, they met inclusion in this review. There were no routinely measured childhood NCDs such as asthma, epilepsy, type 1 diabetes mellitus, or childhood cancers.

## Discussion

This scoping review provides a comprehensive overview of the data landscape for child public health situation, outcome, and impact indicators in FCV settings. A key finding of this review is that a large number of child public health indicators are routinely measured in FCV settings in spite of the contextual challenges. Routinely collected indicators by operational agencies included all four domains of health status, health service, social determinants of health and health behaviours. Most of the indicators that are included in this review were specifically recommended by technical agencies and partnerships, required by donors, and/or used in humanitarian contexts. As such, they are indicators that are recommended or actively measured in places characterised by interruptions in access, subject to short-term funding and/or

**Table 8. Mortality Indicators.**

| Indicator name | Indicator definition/ calculation | Age disaggregation | Category |
|---|---|---|---|
| **Still birth rate (per 1000 births)** | Proportion of all stillbirths (including stillbirths with no foetal heartbeat at admission as well as other stillbirths) among all birth outcomes (x1000) | Newborns | Routinely collected by operational agencies |
| **Stillbirth rate in health facility** | NUMERATOR: Number of fetuses and infants born with no sign of life and born with birthweight of 1000g or more, or after 28 weeks gestation, or 35 cm or more body length in a specified time period; DENOMINATOR: Total number of births (per 1000) at a facility in a specified time period<br>A stillbirth or fetal death is defined as a baby born with no signs of life after a given threshold in health facilities. For international comparison, WHO defines stillbirth as birthweight of 1000 g or more, if the birthweight is not available, a gestational age of 28 weeks or more or a length of 35 cm or more (ICD-10). This indicator should be routinely disaggregated by fresh and mascerated when possible. | Newborns | Recommended for routine collection |
| **Stillbirth rate** | # of fetuses and newborns born after 28 weeks gestation or ≥ 1000g with no sign of life life/ 1000 live births<br>NUMERATOR: # of fetuses and newborns born after 28 weeks gestation or ≥ 1000g with no sign of life<br>DENOMINATOR: 1000 live births | Stillbirth >28 weeks gestation or ≥ 1000g | Recommended for routine collection |
| **Stillbirth Rate** | Number of stillbirths/ Number of live births and stillbirths x 1000. Report in/1000 total births/ month. Stillbirth is defined as a fetal death after 22 weeks of gestation and prior to delivery | None | Recommended for routine collection |
| **Neonatal admission mortality rate (%) (by weight)** | From WHO Paediatric Quality of Care Indicators<br>Neonatal admission mortality rate (%) (by weight) | WHO Quality of Care recommendation | Routinely collected by operational agencies |
| **Institutional neonatal mortality rate (per 1000 live births in facility)** | From WHO Paediatric Quality of Care Indicators<br>Institutional neonatal mortality rate (per 1000 live births in facility) | WHO Quality of Care recommendation | Routinely collected by operational agencies |
| **Outborn Neonatal case fatality rate (%)** | From WHO Paediatric Quality of Care Indicators<br>Outborn Neonatal case fatality rate (%) | WHO Quality of Care recommendation | Routinely collected by operational agencies |
| **[Agency] facility neonatal mortality rate (per 1000)** | Proportion of all neonate deaths that were deliveries that occurred in an [Agency] facility. | Neonates ≤28 days | Routinely collected by operational agencies |
| **Overall neonatal admission mortality rate (%)** | Proportion of neonatal (< = 28 days) exits (exit from neonatal unit) that were deaths. | Neonates ≤28 days | Routinely collected by operational agencies |
| **Neonatal Mortality Rate (NNMR)** | Total number of deaths for newborns < 28 days of life/ Total number of live births x 1000 | <28 days | Recommended for routine collection |
| **Neonatal mortality rate** | NUMERATOR: Number of newborns who died during the first 28 days (day 0-27) of life in health facilities in a specified time period;<br>DENOMINATOR: Total number of live births (per 1000) in a specified time period | By timing of death (early neonatal death = 0 to 6 days; late neonatal death = 7 to 27 days) | Recommended for routine collection |
| **Neonatal mortality rate** | # of deaths in the first 28 days of life/ 1000 live births<br>NUMERATOR: # of deaths in the first 28 days of life<br>DENOMINATOR: 1000 live births | Early neonatal (<7 days after birth) and late neonatal (8-28 days) | Recommended for routine collection |

*(Continued)*

**Table 8.** (Continued)

| Indicator name | Indicator definition/ calculation | Age disaggregation | Category |
|---|---|---|---|
| Pre-discharge neonatal mortality rate | NUMERATOR: Number of babies born live in a facility who die prior to discharge from the facility during the first 28 days (day 0-27) of life in a specified time period; DENOMINATOR: Number of babies born live in a facility in a specified time period | By timing of death (early neonatal death = 0 to 6 days; late neonatal death = 7 to 27 days) | Recommended for routine collection |
| Neonatal cause of death | NUMERATOR: Number of newborn deaths due to<br>• Low birth weight and prematurity<br>• Complications of intrapartum events<br>• Infections (including tetanus, sepsis/meningitis, pneumonia)<br>• Congenital malformations or abnormalities<br>• Other<br>• Unspecified<br>DENOMINATOR: Number of newborn deaths recorded at a facility | By timing of death (early neonatal death = 0 to 6 days; late neonatal death = 7 to 27 days) | Recommended for routine collection |
| Infant Mortality Rate (IMR)/ 1000 live births | Number of deaths among under ones/ Total number of live births x 1000 | <1 year | Recommended for routine collection |
| Under 5 inpatient mortality rate (%) | From WHO Paediatric Quality of Care Indicators Under 5 inpatient mortality rate (%) | WHO Quality of Care recommendation | Routinely collected by operational agencies |
| Under-five mortality | Numerator: Total number of death in children under 5 years during time period Denominator: Total number of children under 5 years multiple by number of days in time period over 10,000 persons | <5 years | Recommended for routine collection |
| Under 5 mortality rate | NUMERATOR: # of deaths of children 1 to 59 months DENOMINATOR: Per 1000 children 1 to 59 months in catchment area | 1-59 months | Recommended for routine collection |
| Under-five mortality rate (1000 population/ month) | Number of deaths among under fives/ mid- period Total under five population x 1000 | <5 years | Recommended for routine collection |
| Under-five crude mortality rate | Not specified | <5 years | Recommended for routine collection |
| Proportion of U5 deaths within 24 hours | Number of under five deaths within 24 hours of admission/ Total number of under five deaths x 100 | <5 years | Recommended for routine collection |
| Adolescent mortality rate | NUMERATOR: # of deaths among adolescents aged 10–19 DENOMINATOR: # of adolescents age 10–19 in the catchment area | 10-19 years | Recommended for routine collection |
| Inpatient mortality rate | Not specified | <5 years and ≥5 years | Routinely collected by operational agencies |
| Timing of death (<24hrs/ >24hrs) | From WHO Paediatric Quality of Care Indicators Timing of death (<24hrs/ >24hrs) | WHO Quality of Care recommendation | Routinely collected by operational agencies |
| Institutional Child Mortality Rate | # of pre-discharge child deaths per 1000 children who visited the health facility NUMERATOR: # of children who died in the health facility before discharge (Includes deaths in the emergency ward but does not include children who died upon arrival at the hospital, child deaths during outpatient visits, and institutional neonatal deaths) DENOMINATOR: # of children who visited the health facility for medical care during reporting period | 0–7 days, 8–27, 28–59 days, 60 days-<1 year, 1-<5 y, 5-<10, 10-<15 y | Recommended for routine collection |
| Inpatient therapeutic feeding centre mortality rate (%) | From WHO Paediatric Quality of Care Indicators Inpatient therapeutic feeding centre mortality rate (%) | WHO Quality of Care recommendation | Routinely collected by operational agencies |

*(Continued)*

**Table 8.** (Continued)

| Indicator name | Indicator definition/ calculation | Age disaggregation | Category |
|---|---|---|---|
| % of children discharged as having died in inpatient treatment for complicated SAM | NUMERATOR: # children discharged as having died in inpatient treatment for complicated SAM<br>DENOMINATOR: # of children discharged from inpatient treatment for complicated SAM | 0-5 mo, 6-23 mo, 24-59 mo | Routinely collected by operational agencies |
| % of children discharged as having died in outpatient treatment for uncomplicated SAM | NUMERATOR: # of children discharged as having died in outpatient treatment for uncomplicated SAM<br>DENOMINATOR: # children discharged from outpatient treatment for uncomplicated SAM | None | Routinely collected by operational agencies |
| Nutrition ITFC Mortality rate (ITFC) | Percentage of cases dead at exit | <5 years and ≥5 years | Routinely collected by operational agencies |
| Ambulatory Therapeutic Feeding Mortality rate (SAM) | Percent of children age 6-59m whose outcome was death after ATFC admission with SAM criteria (includes male/female or no disaggregation, age-groups 6-59m, 6-23m, 24-59m depending on disaggregations used) | 6-59m, 6-23m, 24-59m | Routinely collected by operational agencies |
| Ambulatory Therapeutic Feeding MAM Mortality rate (MAM) | Not specified | <5 years and ≥5 years | Routinely collected by operational agencies |
| % of children discharged as having died in outpatient treatment for MAM | NUMERATOR: # of children discharged as having died in outpatient treatment for MAM<br>DENOMINATOR: # children discharged from outpatient treatment for MAM | None | Routinely collected by operational agencies |
| Death Rate (U5) in Stabilisation Centre (SC) - SAM Treatment | Number of U5 deaths/ Total number of U5 discharges x 100 | <5 years | Recommended for routine collection |
| Death Rate (U5) in Outpatient Therapeutic Programme (OTP) - SAM Treatment | Number of U5 deaths/ Total number of U5 discharges x 100 | <5 years | Recommended for routine collection |
| Death Rate (U5) in Targeted Supplementary Feeding Programme performance Indicators - MAM Treatment | Number of U5 deaths/ Total number of U5 discharges x 100 | <5 years | Recommended for routine collection |
| The proportion of discharges from targeted supplementary feeding programmes who have died, recovered or defaulted | Not specified | Unknown | Recommended for routine collection |
| Proportion of discharges from therapeutic care who have died, recovered or defaulted | Not specified | Unknown | Recommended for routine collection |

*(Continued)*

**Table 8.** (Continued)

| Indicator name | Indicator definition/ calculation | Age disaggregation | Category |
|---|---|---|---|
| **Case fatality ratio** | Numerator: Number of deaths from a specified disease<br>Denominator: Number of cases of a specified disease | Age: <5 years, ≥ 5 years | Required by donors |
| **Case fatality rates for major causes of mortality** | From WHO Paediatric Quality of Care Indicators<br>Case fatality rates for major causes of mortality | WHO Quality of Care recommendation | Routinely collected by operational agencies |
| **Case fatality ratio for all confirmed cases [outbreak disease] admitted for inpatient care at treatment center** | NUMERATOR: # of deaths among cases [outbreak disease] admitted for inpatient care at [agency]-supported treatment centers<br>DENOMINATOR: # of cases [outbreak disease] admitted for inpatient care at [agency]-supported treatment centers | 0-5, 5-24, 25-64, 65+ | Routinely collected by operational agencies |
| **Measles case fatality rate** | The proportion of deaths due to measles within a designated population of cases | <5 years and ≥5 years | Routinely collected by operational agencies |
| **Meningitis case fatality rate** | Proportion of meningitis cases dying from meningitis or its complications in treatment facilities and/or in the community. | <5 years and ≥5 years | Routinely collected by operational agencies |
| **Cholera case fatality rate** | Proportion of cholera cases dying from cholera or its complications in treatment facilities and/or in the community. Standard indicator for adequate case management is a CFR OF >1%. | <5 years and ≥5 years | Routinely collected by operational agencies |
| **Other disease specific outbreak response, specify disease: Case fatality rate** | Not specified | <5 years and ≥5 years | Routinely collected by operational agencies |
| **Mortality rate (TB)** | Not specified | <5 years and 5-15 years | Routinely collected by operational agencies |
| **Visceral Leishmaniasis (Kala Azar) Mortality rate (KA)** | Not specified | <5 years and ≥5 years | Routinely collected by operational agencies |
| **% Patients <15yr dead at 6 months since ART commencement** | Not specified | <5 years and 5-15 years | Routinely collected by operational agencies |
| **% Patients <15yr dead at 12 months since ART commencement** | Not specified | <5 years and 5-15 years | Routinely collected by operational agencies |
| **% Patients <15yr dead at 24 months since ART commencement** | Not specified | <5 years and 5-15 years | Routinely collected by operational agencies |
| **Proportional mortality** | Proportion of deaths attributable to a particular cause among the population<br>Number of deaths due to a particular cause/ Total number of deaths x 100 | Crude and under 5 | Recommended for routine collection |
| **Case Fatality Rate (IPD)** | Number of deaths/ Number of admissions due to a particular cause x 100 | Crude and under 5 | Recommended for routine collection |

*(Continued)*

**Table 8.** (Continued)

| Indicator name | Indicator definition/ calculation | Age disaggregation | Category |
|---|---|---|---|
| **Case fatality rate for pertussis, cholera, meningitis, hepatitis E, diphtheria, dengue** | Not specified | Pertussis disaggregated to age 1-4 years | Recommended for routine collection |
| **In-hospital paediatric case fatality rate** (*by common paediatric conditions*) | % of children who were diagnosed with Sepsis, Pneumonia, Malaria, Meningitis or Severe Acute Malnutrition (SAM) and died in the health facility<br>NUMERATOR: % of children who were diagnosed with Sepsis, Pneumonia, Malaria, Meningitis or SAM and died in the health facility (*Includes deaths in the emergency ward but does not include children who died upon arrival at the hospital, child deaths during outpatient visits, and institutional neonatal deaths*)<br>DENOMINATOR: # of children who visited the health facility and were diagnosed with Sepsis, Pneumonia, Malaria, Meningitis or SAM during reporting period | 0–7 days, 8–27, 28–59 days, 60 days- < 1 year, 1- < 5 y, 5- < 10, 10- < 15 y | Recommended for routine collection |
| **TB case fatality rate** | Number of TB deaths/ total number of TB cases notified for treatment x 100 | 0-4, 5-14, 15-17, 18 and above | Recommended for routine collection |
| **Mortality rate in surgical patients** | Total deaths amongst inpatients who had 'went to operating theatre' is ticked in their inpatient admission record. Surgical mortality rate should include any death occurring within 30 days or surgery or during the same admission. | <5 years and ≥5 years | Routinely collected by operational agencies |
| **Suicide rate, disaggregated** | NUMERATOR: # of suicide deaths in a year, disaggregated by age and sex<br>DENOMINATOR: Per 100,000 patients in the catchment area | Unknown | Recommended for routine collection |
| **Gender-specific Mortality Rate** | Number of male (or female) deaths within specified age group/ Population within same age group x 1000 | Crude, under 5, and infant | Recommended for routine collection |
| **Proportion of female deaths that occurred due to gender-based causes** | The number of women or girls who were murdered during a specific time period (e.g., the past 12 months) for gender-based reasons. These reasons include dowry death, family honor, IPV, murder with rape, killings of prostitutes, female infanticide, and other deaths where reports confirm that the deaths occurred as a result of women or girls being targeted on the basis of gender (for example, a serial killer who has singled out women as victims).<br>NUMERATOR: The number of women or girls who were killed for gender-based reasons during a specific time period (e.g., the past 12 months)<br>DENOMINATOR: The total number of women or girls murdered during the same time period | None | Recommended for routine collection |
| **Excess female infant mortality (sex ratios up to age 1 year)** | The number of females per 100 males at two points in time — age 0-1 for excess infant mortality, and age 0-4 years for excess child mortality.<br>NUMERATOR: Number of females aged 0-1 year at a specified point in time<br>DENOMINATOR: Number of males aged 0-1 year at the same specified point in time. | 0-1 year | Recommended for routine collection |
| **Excess female child mortality (sex ratios under 5)** | The number of females per 100 males at two points in time — age 0-1 for excess infant mortality, and age 0-4 years for excess child mortality.<br>NUMERATOR: Number of females aged 0-4 years at a specified point in time.<br>DENOMINATOR: Number of males aged 0-4 years at the same specified point in time. | 0-4 years | Recommended for routine collection |
| **% of child deaths with mortality review & documentation of lessons learned** | % of child deaths with mortality review & documentation of lessons learned | WHO Quality of Care recommendation | Routinely collected by operational agencies |

interrupted funding for interventions, and subject to logistical and technical challenges that relate to the specific crisis and setting.

The sheer number of indicators that are recommended for routine collection by technical agencies and partnerships is striking. Given the paucity of routinely available data on child public health worldwide and the well-described challenges for data collection in FCV settings [9,20] it is concerning that over 500 indicators are recommended by major supporting agencies and partnerships for routine collection. It is important to note that the indicators

included in this review are limited to situation, outcome and impact indicators. Output and process indicators, which often form the bulk of reporting by operational agencies [9], were not included in this review. The very large number of recommended indicators would require a significant amount of human capital, financial resources, and infrastructure to measure, analyse and interpret. It is not realistically achievable by governments and nongovernmental organisations working in FCV settings and it suggests competing priorities between sectors and limited coordination of activities across sectors.

The current collection of indicators is also not appropriate, as many indicators overlap, and some are similar but not exactly the same. It not clear whether all of the recommended indicators would be useful for decision-making by operational actors. It is noteworthy that specific sectors, such as nutrition and child protection, recommended particularly large numbers of indicators (Table 7), suggesting a lack of prioritisation of topics by actors working within the respective sectors as well as lack of coordination and collaboration for prioritisation, standardisation, and harmonisation of indicators across sectors and agencies. The use of intersectoral indicators would help to both reduce the number of indicators and can provide a more human-centred overview of child public health. Finally, the large number of recommended indicators may be confusing for monitoring and evaluation teams, and likely reinforces variations in measurement. Heterogeneity in measurement by teams and agencies reduces the reliability and validity of the data and makes pooling of data challenging. As such, measurement variability hampers meaningful accountability to populations, by limiting our ability to understand whether an intervention was safe, appropriate, and effective for the target population. Data practices that hamper accountability serve to reinforce a colonial approach to aid, where the outcomes and impacts of interventions on the affected population are not prioritised.

Establishment of consensus about priority topics across sectors that address child public health, alongside standardisation of prioritised indicators could both drastically reduce the number of recommended indicators and simultaneously improve indicator quality, and ultimately, accountability to affected populations. Indicator prioritisation should consider usefulness for operational decision-making in the dynamic environment of FCV and humanitarian settings. Prioritisation of indicators may be most effective if based upon decision-making needs.

It is important to consider that standardisation of indicators may not be feasible for each type of data source. Child public health data are collected in FCV settings by a range of methods, such as population-based surveys, review of clinical information, facility assessments [21], rapid needs assessments, qualitative interviews with facility staff, patients and/or affected populations, and quality of service assessments. Each type of data source contains different information that may be used for baseline data, monitoring changes in population needs, and/or evaluating the impact of interventions. However, definitions of both the numerator and denominator may differ by data source. For example, the denominator in a population-based survey, a census, or the quantification of a population in need quantified by UN OCHA, are likely to differ substantially. The accuracy of denominators may be uncertain and difficult to assess in FCV settings, which are often characterised by weak vital registration systems, fluid population movement, displacement of populations across administrative or even national borders, barriers in access to the affected population, and lack of expertise and resources needed to estimate population size [22]. Some of the indicators included in this review either lacked a definition for the denominator entirely or provided a definition that required significant interpretation. Some indicators suggested multiple, differing data sources for denominators. These practices limit the reliability of indicator measurement and preclude pooling of data. Efforts to establish consensus on a shortlist of priority child public health indicators for routine measurement in FCV settings must therefore consider the types of data sources that will be used and develop clear definitions of the numerator and denominator.

It is encouraging that 152 indicators were feasible for five agencies to routinely collect despite the contextual challenges in FCV settings (Table 7). It is important to reiterate that these indicators are actively measured by five large operational nongovernmental agencies working in FCV settings (S4 Text) and is likely to differ from the kinds of measurements made by smaller organisations. However, the finding indicates that the measurement of child public health outcome, impact and situation indicators is feasible despite the significant and changeable contextual challenges. This lends further support to the argument for the need to develop adaptations of "gold standard" outcome indicators for measurement in settings where the highest standard of measurement is not possible to consistently collect. In the absence of existing routine data on child public health in FCV settings, researchers and the public health community have developed methods to create estimates based on exiting data, such as data from the Demographic and Health Surveys [23, 24]. Our findings suggest that in many settings, reliance on estimates is not necessary – that it is possible to collect data in these settings. There are notable exceptions, in contexts where health systems have been severely destroyed and/or access is restricted, such as the current situations in Gaza and Sudan, respectively [25, 26]. In such contexts, the use of adapted indicators for proxy measures is a realistic solution that could enable ongoing routine measurement of outcomes. In addition to the creation of adapted indicators, there is a need for global consensus on a shortlist of prioritised indicators for routine measurement across agencies and contexts, and a system to support the pooling and sharing of these data.

During the course of this review, a number of indicator banks were also identified, some containing hundreds of indicators for a range of sectors and topics [27, 28]. Indicators from these banks were not included in the review because they were not recommended for routine collection across FCV settings but serve as a resource of potential measures for actors, often with specific focus areas, such as infectious disease outbreak. A shortlist of priority indicators feasible for routine measurement in FCV settings with specific adapted proxy indicators, supplemented by a bank of high-quality indicators for routine measurements may be a preferable alternative to the current longlists.

It is also encouraging that only two of the six approached donors, USAID BHA and ECHO, had requirements for the measurement of child public health; the other donors advised that they do not have specific requirements, but that operational teams are asked to propose measures for reporting. Interestingly, one donor agency responded to our enquiry with a request for recommendations about child public health outcome indicators that they should consider requiring. Donor reporting requirements are often cited as the main driver for indicators measured by operational teams [29, 30]. The findings from this review suggest that operational agencies and child public health professionals have an opportunity to influence donor requirements and expectations for data collection. To this end, a core list of child public health indicators for measurement in FCV settings would be of value.

If a core list of child public health indicators were routinely measured by all actors, including ministry, nongovernmental organisations, and other operational groups, and if these data were reliably measured, we could understand the corresponding child public health status and needs of children in a given crisis context and globally. For the first time, we would be able to base our response to crises on actual data on child public health, rather than estimates.

With over 500 indicators recommended for routine collection, it is unsurprising that there are overlaps and similarities in recommended indicators from technical organisations and partnerships. While a detailed assessment of quality was beyond the scope of this review, the included indicators were assessed for the degree of interpretation required in order to measure them. Less than half of all indicators were of sufficient clarity that they can be measured without some degree of interpretation (Table 5. Included indicators by category). Given the

vast number of indicators recommended by technical agencies, partnerships and academics, it is concerning that more than half of them required either some or significant interpretation to measure. Several of the recommended indicators contained typos in the definition or title that led to discrepancies between the title and the definition and led to uncertainty in what was meant to be measured. While typos are unfortunate, they are easy to correct. A more challenging correction would be the development of consensus on a shortlist of priority child public health outcome, situation, and impact indicators for measurement in FCV settings. The latter is essential if we are to hold ourselves accountable to children and populations affected by crisis.

A similar trend in was observed in the routinely collected indicators, with just under half requiring at least some interpretation. When we enquired about definitions from one agency, we were informed that indicators were deliberately not defined and that country teams were encouraged to use "common sense". The failure to define what is being measured renders the entire measurement process unreliable. This relaxed approach to tracking routine indicators suggests a systemically colonial approach to crisis response without accountability for the resulting outcomes from the services that are provided.

Age disaggregation practices and recommendations vary widely and were even found to vary among indicators recommended or used by the same organisation. There were no clear trends from any sector, organisation, or donor, except that nutrition indicators are most often recommended for measurement in children aged 6-59 months. Even the indicators for infant and young child feeding used variable age disaggregation, all of them measuring some group(s) under 2 years of age, but with significant variations in the definition of these groups. Diaz et al have published a strongly argued recommendation for standardised age disaggregation [31]; widespread adoption of standardised age disaggregation across all public health programming in FCV settings would improve the ability of governmental, nongovernmental, and academic institutions to collect, compare, share, and pool data.

The only NCD measures routinely collected by operational teams focused on diseases that are more commonly seen in adults. The lack of child specific NCD indicators routinely measured suggests that the decisions for what to measure are made by individuals or teams that have limited child public health knowledge. With children accounting for a significant proportion of the population in crisis-affected areas - and an even higher proportion of health care contacts in humanitarian settings [32] - a lack of child public health expertise among international NGOs is unacceptable. The findings of this review are consistent with other papers that have identified a lack of child health expertise in humanitarian settings more generally, and a need to train our colleagues who are tasked with providing care and designing and delivering interventions to support children in crisis contexts [33].

Sexual and gender-based violence (SGBV) indicators tended to be recommended by a range of different sectors, indicating a broader recognition across sectors of the relevance of SGBV to child public health. Of the SGBV indicators, mental health and psychosocial support (MHPSS) and access to care were the main areas of focus. Agencies routinely collecting MHPSS indicators tended to focus on SGBV and on change/improvement in mental health outcomes after service provision. Recommended indicators explored MPHSS service needs, outcomes, and referral pathway functioning, as well as caregiver and teacher MHPSS needs and support. The inclusion of indicators on caregiver and teacher MHPSS among recommended indicators is encouraging, as it indicates a recognition of caregiver mental health as a major determinant of child health, development, and safety.

The lack of indicators measuring early child development (ECD) and disability indicates a major technical gap in child public health expertise by operational agencies, donors, and technical organisations supporting programmes in FCV settings. There were a small number

of indicators on injury (N = 5) and disability (N = 9) recommended by child protection actors. No measures of ECD were identified by the review. Early child development and disability are both markers of child health and also determinants of child health and protection; they have significant impacts on child health and children's risk of violence and exploitation throughout childhood, adolescence, and adulthood [34, 35]. The cumulative harmful effect of early childhood adversity on physiological, neurological, and socioemotional development are well described [4]. As such, developmental status and disability have important implications on child health and child public health needs and outcomes in all settings, and especially in FCV settings where children are at higher risk of experiencing multiple forms of severe adversity. For example, it can be argued that children with severe acute malnutrition (SAM) have experienced multiple forms of severe adversity that resulted in them developing SAM. It is well documented that children with direct exposure to armed conflict experience a range of severe forms of adversity including witnessing and/or being subjected to physical, psychological and emotional violence, lack of access to basic needs, loss of one or both parents, forced displacement, and early marriage, to name just a few. These children are also known to be at significant risk of disabling injuries from combat activities, including blast injuries, penetrating wounds including penetrating head injuries from bullets, shrapnel and other objects, crush injuries, and burns [36]. Children can also be injured by combat activities long after hostilities have ended; up to 50% of injuries from unexploded ordnance are in children, sometimes decades after the end of the conflict [37]. These are just a few examples of why ECD and disability are important markers of child health and child public health. It is concerning that ECD and disability are not prioritized in measurement of child public health outcomes in FCV settings. Given the pervasive and potentially profound impacts of crises on child development and disability, the consequences of failing to measure ECD and childhood disability are significant. At best, programmes and funding decisions will be inadequately informed. Prioritisation of the use of resources may be inappropriate or may fail to address the needs of children. At worst, ill-informed programme design and implementation may reinforce existing inequalities, exacerbate health harms to children, or cause further harm.

## Limitations

This review compiled recommended indicators from large organisations and partnerships that provide technical guidance and recommendations for operational agencies. The literature review was restricted to English language publications. Similarly, the routinely collected indicators were from a handful of large international nongovernmental organizations, and reporting requirements were sought from six large government and private donors that fund interventions in humanitarian contexts. As such, the indicators included in this review are by no means an exhaustive list; they are more representative of large and well-funded institutions and partnerships that are primarily based in Europe and North America and support and/or operate in FCV settings.

Inclusion of qualitative indicators was outside of the scope of this review. Both quantitative and qualitative data are necessary to support safe, relevant, and effective program decision-making. Further research on qualitative child public health measures recommended, required, and actively measured in FCV settings would be helpful and provide further depth to our understanding of child public health data practices, gaps, and needs in FCV settings.

The paucity of ECD and disability indicators measured by operational teams may be in part due to bias from agency contacts who shared indicators with the research team. The lists of routinely collected indicators were provided primarily by health and nutrition staff; our sample may be missing some non-health or nutrition indicators that are routinely collected. During data collection, the study team provided written definitions of child public health and

the domains included in this review, as well as verbal explanations to participating agencies. If the shared indicators were focused on child health and nutrition, follow up was made to enquire about other indicators to measure child public health. However, the siloed nature of aid work may have impacted the kinds of indicators we received. It should be noted that three of the five organisations that shared their routinely collected indicators included education, protection and/or MHPSS indicators. If there is a gap in our dataset for child public health indicators not strictly falling into health or nutrition, it would suggest that public health workers may not recognise the child public health work done by teams in other sectors outside of health and nutrition and/or the value of the data they are collecting, and a more general lack of child public health expertise.

## Conclusions and recommendations

The routine measurement of child public health indicators in fragile, conflict-affected, and vulnerable settings is not only possible, but it is already being achieved. Measurement of child public health in FCV settings is characterised by an excessive number of recommended indicators, duplication, poor definitions, and siloed sector-specific focus. Improving the measurement of child public health in FCV settings requires recognition of the child public health work done by actors in sectors other than health and nutrition. While there are recommended and routinely collected indicators on social determinants of child health, important gaps remain. The most conspicuous gaps include early child development and disability, which are both child public health outcomes and also determinants of child health, wellbeing, safety, and life trajectory. There is a need to establish consensus on priority child public health topics for measurement and a shortlist of high quality, appropriate and relevant measurements that governmental and nongovernmental actors can be reasonably expected to measure routinely in FCV settings. The prioritisation of indicators should be based on decision-making needs in a dynamic context. Proxy indicators should be developed for periods where routine measurement is hampered. Clearly defined indicators and consensus on age disaggregation would enable pooling and comparison of data that could inform an evidence base for data-driven decision-making in FCV settings. Child public health expertise and data are not a "nice to have" in FCV settings, where a significant proportion of the population are children; they are essential for appropriate and relevant measurement of child public health, guidance on interventions, evaluation of interventions, and accountability to affected populations.

## Supporting information

**S1 Text. Study protocol.**
(DOCX)

**S2 Text. Codebook with definitions.**
(XLSX)

**S3 Text. Preferred Reporting Items for Systematic reviews and Meta-Analyses extension for Scoping Reviews (PRISMA-ScR) Checklist.**
(DOCX)

**S4 Text. Sources of included indicators.**
(DOCX)

**S1 Fig. Health domains.**
(TIF)

**S1 Data. Complete list of included indicators.**
(XLSX)

## Acknowledgments

The authors would like to thank Bethany Tapster at the Leeds Teaching Hospitals NHS Trust for support in developing the database searches. We would also like to thank the organisations that spoke to us and shared their indicators.

## Author contributions

**Conceptualization:** Ayesha Kadir.

**Data curation:** Ayesha Kadir, Amy J. Stevens, Sham Lal.

**Formal analysis:** Ayesha Kadir.

**Investigation:** Ayesha Kadir, Amy J. Stevens, Emi A. Takahashi.

**Methodology:** Ayesha Kadir, Amy J. Stevens, Emi A. Takahashi, Sham Lal.

**Project administration:** Ayesha Kadir.

**Supervision:** Ayesha Kadir.

**Visualization:** Ayesha Kadir.

**Writing – original draft:** Ayesha Kadir.

**Writing – review & editing:** Amy J. Stevens, Emi A. Takahashi, Sham Lal.

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
