## [Decision Letter · Decision Letter 0]

26 Aug 2024

PGPH-D-24-01333

Child public health indicators for fragile, conflict-affected, and vulnerable settings: a scoping review

Dear Dr. Lal,

Thank you for submitting your manuscript to PLOS Global Public Health. After careful consideration, we feel that it has merit but does not fully meet PLOS Global Public Health’s publication criteria as it currently stands. Therefore, we invite you to submit a revised version of the manuscript that addresses the points raised during the review process.

We look forward to receiving your revised manuscript.

Kind regards,

Hannah Tappis, DrPH, MPH

Academic Editor

Journal Requirements:

Additional Editor Comments (if provided):

Please define and distinguish between 'humanitarian', 'fragile', 'conflict-affected' and 'vulnerable' settings or explain why these terms are used interchangeably. Table 1 cites a broad WHO definition for FCV, but does not explain whether this is an objective or subjective classification or how it was applied in search and screening criteria. Is there a list of countries or sub-national contexts that are considered FCV at any given time? Were articles included only if authors used these exact terms to describe settings? Or if settings of interest met certain criteria? Further explanation will both enhance the replicability of the review, and guide readers in interpretation of recommended indicators and measurement methods.

Reviewers' comments:

Reviewer's Responses to Questions

**Comments to the Author**

1. Does this manuscript meet PLOS Global Public Health’s publication criteria ? Is the manuscript technically sound, and do the data support the conclusions? The manuscript must describe methodologically and ethically rigorous research with conclusions that are appropriately drawn based on the data presented.

Reviewer #1: Partly

Reviewer #2: Yes

2. Has the statistical analysis been performed appropriately and rigorously?

Reviewer #1: N/A

Reviewer #2: N/A

3. Have the authors made all data underlying the findings in their manuscript fully available (please refer to the Data Availability Statement at the start of the manuscript PDF file)?

Reviewer #1: Yes

Reviewer #2: Yes

4. Is the manuscript presented in an intelligible fashion and written in standard English?

Reviewer #1: Yes

Reviewer #2: Yes

5. Review Comments to the Author

Reviewer #1: The author addresses a relevant topic in public health highlighting the gaps in children's public health data in fragile, conflict-affected, and vulnerable settings (FCV). They provided in their introduction, a comprehensive overview of the challenges faced by children during FCV and the need for more standardized children's public health indicators. However, it could benefit from clearer structure and concision. They presented many relevant findings determining trends and gaps in the current system. Providing more actionable recommendations would enhance that work. The article is comprehensive overall, but would benefit of more concision and organization.

Major recommendations

Abstract

Conclusion

- Findings Implications: In the Conclusions section, the first sentence seemed to be weak. Consider highlighting the implications of your findings. For example, discuss the feasibility of measuring child public health outcomes in FCV settings despite challenges, and its relevance. The practical implications of the excessive number and poor definition of indicators recommended by technical experts.

- Recommendations: The recommendation may be more concise. In addition to the consensus, consider emphasizing needs for standardization, prioritization and adressing the gaps identified in the study. An final sentence about the finality of that recommendation would be good like : to improve monitoring and evaluation in these settings or improve child healh during FCV or....

Background

While the introduction covers a broad spectrum of challenges and contexts, ensure that the narrative consistently emphasizes the public health aspects relevant to children in FCV settings.The introduction highlighted the problem of humanitarian data limitation and most importantly the mismatch between publicly available data and tons of internal data collected by NGOs. The paragraph after explained the lack of standardization of those data. The authors may consider presenting those arguments in more integrated and concise ways to emphasize their importance.

There is a lack of connection between the idea expressed in the sentence from lines 66 to 68 and the one in lines 70-71. A better transition can be useful here. A better transition can also be done between sentences 84-86 and 88-89

Make sure that each point stays connected back to the specific context of child public health in FCV settings and how the lack of data impacts public health responses. This ensures that the introduction remains focused on the study's relevance.

Methods

The Methods section contains important details but lacks clear organization and structure. It would benefit from separating different stages of the review process into distinct paragraphs or subsections, such as search strategy, screening process, inclusion criteria, and data analysis. It would be good as well to present a specific paragraph about the grey literature assessment. Were there pre-approved strategies to resolve disagreements among reviewers, if yes, specify it.

The authors specified that the DuckDuckGo search 153 was run on 14 March 2023. Is there a specific reason to provide a date only for that activity and not for the others? It would be better to know the general timeframe of the whole data collection process (from …. To …. ), instead of a single date for 1 activity.

Mention how inter-rater reliability was ensured during the screening and inclusion process. This could include details on pilot testing of the inclusion criteria and coding framework.

Results

Adding the proportion to table 6, as it is done for table 7, will provide more insights.

Discussion

The first paragraph of the discussion could synthesize better the scoping review's main findings.

Second paragraph of the discussion. More than 500 indicators are recommended. The authors can explain this more striking finding and its relevance. Do all those 500 indicators seem necessary? The authors mentioned they are not realistic for governments: specifying why would reinforce the statement (assess the 500 will require more funding, staffing, logistics, will not be sustainable … ? )

While the section discusses the large number of recommended indicators and feasibility issues, it could delve deeper into interpreting why these challenges exist. For instance, explore the underlying reasons for the discrepancy between recommended indicators and feasible measurements in FCV settings.

Consider strengthening the discussion of gaps in measuring early child development and disability by exploring the potential consequences of these gaps on program effectiveness and child health outcomes.

Conclusion and recommendation

The conclusion outlines key issues but lacks a clear pathway for addressing them. Instead of stating findings already explained in results and discussions, it could focus more on presenting a succinct summary of the implications of the findings for policy, practice, and future research and highlight actionable steps that stakeholders can take to address the identified challenges in measuring child public health in FCV settings.

Minor recommendations

Ensure the abstract is concise and uses precise language.

Line 46: instead of "proving this is possible" in the Conclusions section, you might say "demonstrating feasibility."

Some sentences could be streamlined for clarity and conciseness. For example, sentences like "In the absence of routine child public health data, humanitarian actors delivering public health services in these settings often turn to survey data..." could be broken down into shorter, clearer segments for easier comprehension.

Editing: conduct a thorough final reading for missing words or grammatical errors. For example, in Table 3. Correct impact indictors to impact indicators.

Reviewer #2: A very useful review that presents a clear need for further standardization and alignment of child health indicators in FCV settings.

The recommendations can be improved by considering the following:

Standardization may only be possible ‘within’ the data source type such as the following:

-Population-based household survey (from DHS down to local rapid household survey);

-Clinical information system such as paper logbooks or registers up to electronic registers or medical records (often called health management information system or routine health information system).

-Health facility assessment such as HERAMS, WHO Harmonized Health Facility, or rapid/local health facility assessments, which may include checks on availability of human resources and essential drugs/equipment, interviews with patients and health workers up to observations of care against clinical guidelines.

-Others (e.g., review of charts, clinical notes, other program records)

-As an example, the WHO quality of care indicator, “% sick children 2months – 5years assessed on IMCI Criteria and appropriately treated” is most appropriately assessed using observation by a independent clinical assessor when a guidelines-based electronic medical record is not available (e.g., WHO digital adaptation kit for child health in emergencies).

Recommendations could provide a standard definition by data type. For example, coverage of measles immunization estimated by a household survey will be defined differently than coverage of measles immunization estimated through the clinical information system. One reason is that denominators will be defined differently or come from different information systems. Surveys provide their own denominator while clinical information systems rely on other sources such as OCHA for population in need or a separate enumeration or other projection. And often, the numerators are also obtained differently (e.g., mother’s report, home vaccination card, marker on child’s finger, or tally of vaccines distributed).

A more general discussion of denominator issues in FCV settings, including how availability of denominators varies by data source, concerns about the accuracy of denominators used in clinical information systems even in more stable settings, etc. and its affect on standardization.

6. PLOS authors have the option to publish the peer review history of their article (what does this mean? ). If published, this will include your full peer review and any attached files.

**Do you want your identity to be public for this peer review?** For information about this choice, including consent withdrawal, please see our Privacy Policy .

Reviewer #1: No

Reviewer #2: No

---

## [Editor Report · Decision Letter 1]

24 Sep 2024

PGPH-D-24-01333R1

Child public health indicators for fragile, conflict-affected, and vulnerable settings: a scoping review

Dear Dr. Lal,

Thank you for submitting your revised manuscript to PLOS Global Public Health. It appears that the point-by-point response to reviewer comments was not included in the resubmission package. Only responses to three editorial comments was uploaded. 

Please submit your revised manuscript, along with a point-by-point response to all reviewer feedback by Oct 24 2024 11:59PM. If you will need more time than this to complete your revisions, please reply to this message or contact the journal office at globalpubhealth@plos.org. Please include the following items when submitting your revised manuscript:

We look forward to receiving your revised manuscript.

Kind regards,

Hannah Tappis, DrPH, MPH

Academic Editor
---

## [Editor Report · Decision Letter 2]

27 Sep 2024

Child public health indicators for fragile, conflict-affected, and vulnerable settings: a scoping review

PGPH-D-24-01333R2

Dear Mr Lal,

We are pleased to inform you that your manuscript 'Child public health indicators for fragile, conflict-affected, and vulnerable settings: a scoping review' has been provisionally accepted for publication in PLOS Global Public Health.

Best regards,

Hannah Tappis, DrPH, MPH

Academic Editor